# What Is Left for Real-Life Lactate Monitoring? Current Advances in Electrochemical Lactate (Bio)Sensors for Agrifood and Biomedical Applications

**DOI:** 10.3390/bios12110919

**Published:** 2022-10-25

**Authors:** Juan José García-Guzmán, Alfonso Sierra-Padilla, José María Palacios-Santander, Juan Jesús Fernández-Alba, Carmen González Macías, Laura Cubillana-Aguilera

**Affiliations:** 1Instituto de Investigación e Innovación Biomédica de Cadiz (INiBICA), Hospital Universitario ‘Puerta del Mar’, Universidad de Cadiz, 11009 Cadiz, Spain; 2Department of Analytical Chemistry, Institute of Research on Electron Microscopy and Materials (IMEYMAT), Faculty of Sciences, Campus de Excelencia Internacional del Mar (CEIMAR), University of Cadiz, Campus Universitario de Puerto Real, Polígono del Río San Pedro S/N, Puerto Real, 11510 Cadiz, Spain; 3Department of Obstetrics and Gynecology, Hospital Universitario de Puerto Real, Puerto Real, 11510 Cadiz, Spain

**Keywords:** lactate monitoring, electrochemical (bio)sensors, agrifood control, biomedicine, sports medicine, e-health

## Abstract

Monitoring of lactate is spreading from the evident clinical environment, where its role as a biomarker is notorious, to the agrifood ambit as well. In the former, lactate concentration can serve as a useful indicator of several diseases (e.g., tumour development and lactic acidosis) and a relevant value in sports performance for athletes, among others. In the latter, the spotlight is placed on the food control, bringing to the table meaningful information such as decaying product detection and stress monitoring of species. No matter what purpose is involved, electrochemical (bio)sensors stand as a solid and suitable choice. However, for the time being, this statement seems to be true only for discrete measurements. The reality exposes that real and continuous lactate monitoring is still a troublesome goal. In this review, a critical overview of electrochemical lactate (bio)sensors for clinical and agrifood situations is performed. Additionally, the transduction possibilities and different sensor designs approaches are also discussed. The main aim is to reflect the current state of the art and to indicate relevant advances (and bottlenecks) to keep in mind for further development and the final achievement of this highly worthy objective.

## 1. Introduction

Within the spectrum of relevant biomarkers, L-lactate is rising as one of the most significant due to its intrinsic role in diverse real-life scenarios. As is well-known, lactate is a metabolite that originates from anaerobic metabolism. This metabolic route is triggered when the regular aerobic metabolism is not enough to satisfy the energy requirements. Typically, an extra energy demand occurs during physical training, increasing the lactate level in blood from 0.5–2.0 to 25 mM or even more in intense exercise. Interestingly, the generation of lactate also involves a proton concentration increment and the consequent pH diminishment, leading to cell acidosis. Moreover, the accumulation of lactate in working muscles during the exercise is conducive to diverse health issues from slight fatigue to severe pain. This is why real lactate monitoring is frequently pursued in sport medicine [1,2]. Particularly, the term lactate threshold was coined in this ambit to draw the highest effort limit for an athlete with no risk of lactate accumulation complications. However, the most common practices to determine lactate pass through blood extraction, which limits its continuous determination and hinders an easy and comfortable assessment [3]. For these reasons, a reliable and robust lactate (bio)sensor would provide a game changer tool for physicians. Optimal trainings may be designed in order to minimise the risk for the athlete avoiding injuries and granting the maximum reward.

Nevertheless, the importance of lactate in the agrifood field is also vast and very essential to the educated observer. In the first place, lactate is frequently used as an acidulant additive (E270) due to its nonvolatile properties. More importantly, derived from its anaerobic natural origin, lactate can be found in fermented products such as milk, yoghurt, cheese, etc. Additionally, other more exotic foodstuffs (sauerkraut and kimchi) can contain it as well. It is not difficult to reason that lactate concentration may be a reliable indicator of the fermentation process and the quality of the prepared foodstuff. It is also very significant in the assessment of beverage spoilage. For instance, juice degradation can be evaluated by employing lactate concentration to estimate lactate-consumer bacteria present in the drink. The uncontrolled proliferation of microorganisms will clearly impoverish the quality of the juice, precluding its regular consumption. Remarkably, lactate has an essential role in the wine industry due to its inverse relationship with malic acid during wine fermentation, being also an indicator of its organoleptic properties. Nonetheless, it is noteworthy to mention that in this scenario, D-lactate isomer generation is considered as a symptom of beverage spoilage. Furthermore, D-lactate is also recurrent evidence of decomposition in solid food such as meat or egg [4]. In addition, lactate determination is relevant in other very raw food, namely farmed fish used in aquaculture. Frequently, fish tanks are overpopulated with specimens, provoking high levels of stress, which results in an increase in lactate and diseases. Therefore, real lactate monitoring would assist in paving the way for more rightful breeding practices in this flourishing industry [5].

Obviously not only is lactate interest found in fish diseases, but also in human wellbeing. Firstly, several diseases increase regular blood lactate leading to ischemic situation [6]. Therefore, the inverse is also true; it is demonstrates that lactate is a potential biomarker in the early diagnosis and monitoring of several diseases (e.g., diabetes or cardiac issues) [7]. In the same vein, lactate assessment is highly appreciated in intensive care units (ICUs), where an abnormal level is a beacon of meaningful information of the current and future status of the patient. In this sense, it is possible to relate it with situations such as haemorrhagic shock, pulmonary embolism, cardiogenic shock, respiratory poisoning and renal failure, among others [8,9,10,11,12,13]. Moreover, alternative studies pointed out the existence of abnormal lactate concentration in cancer cells during metastases [14]. Moreover, lactate is also relevant in neuroscience, being a sleep biomarker and an indicator of brain metabolism [15,16]. In addition, as has been previously mentioned, lactate is inversely related with pH; hence, a high increase of lactate causes lactic acidosis, altering the acid-base homeostasis and negatively affecting the entire organism if the pancreas and liver are not able to excrete it properly [17]. These two parameters, lactate and pH, can be extremely important in certain scenarios where the person is weakened or under a specific risk. For instance, during delivery, it is highly recommended to monitor it in order to distinguish between respiratory acidosis and metabolic acidosis, thus helping to prevent the sequelae of lactic acidosis in the newborn. Unfortunately, this is not feasible yet, mainly due to the current methodology designed for this purpose, which is bloody and complex. The extraction of blood is, at the present time, unavoidable and is very limited during births extended in time, for evident reasons. Ironically, it is in the difficult births when the risk is higher and the lactate/pH monitoring is of paramount importance due to the dangers for the baby being critical (e.g., postnatal neurological problems, asphyxia, premature death, etc.) [18]. Furthermore, in the reproductive field, it has also been demonstrated that lactate is a relevant species for cell cultures needed in embryonic cell growth, especially in the first stages [19].

At this point, the reader is able now to assimilate the importance of lactate and the real need to determine it in both agrifood and biomedicine environments (Figure 1).

In this sense, the scientific community has invested huge efforts and funds to achieve this. Time has not been wasted and the initial methodology proposed by Barker and Summerson in 1941 based on non-enzymatic colorimetric assays is now considered obsolete [20]. More specific, sensitive and reliable procedures have taken up the torch. Regarding the biomedicine ambit, high-performance liquid chromatography (HPLC) [21], fluorometry [22], chemiluminescence [23], microwave sensing [24], holography [25] and magnetic resonance spectroscopy [26] have been already employed to determine lactate. It is noteworthy to mention that lactate determination in ICUs and clinics is currently carried out by using an enzyme-based spectrophotometric and colorimetric method prior to blood sampling. Usually, lactate dehydrogenase is employed as biological recognition element to catalyse lactate–pyruvate conversion through NADH formation. Subsequently, NADH can be determined via spectrophotometric techniques and can find out the initial lactate concentration in the fluid [27]. Despite the suitable sensitivity and reliability supplied by this technique, it only allows for discrete measurements in time, requires the addition of an additional reagent (NAD^+^) and more importantly, it is not possible to have an unlimited blood sampling due to obvious reasons. On the other hand, spectrophotometry [28], electrophoresis [29], nuclear magnetic resonance spectroscopy [30], electrochemiluminescence [31], high-performance liquid chromatography [32] and liquid chromatography–mass spectrometry [33] have also been applied in food analysis. However, despite the excellent outputs that may be provided, all these methodologies also possess several important drawbacks such as complex sample treatments, highly trained personnel requirements, slow response time and in some cases, expensive instruments. Furthermore, other aspect should be also evaluated: (i) the possibility of online analyses, (ii) its performance in ICUs/clinical environment and (iii) feasible real-time monitoring. In our opinion, these final features are the key to evolve from conventional lactate determination to a meaningful and profitable analysis. In this sense, one of the best alternatives to reach the required milestones can be found in electrochemical (bio)sensors.

Electrochemical devices have a wide range of advantages such as simplicity, rapid response, high sensitivity, low cost and direct measurement with low or no sample preparation. In addition, portability and the possibility of in situ and online analysis in a minimally invasive manner are also among their most important characteristics [17,34]. Particularly, selectivity can be greatly enhanced by using a biological recognition element within the body of the sensor. In the case of lactate, enzymes are frequently used due to their biocatalyst properties and high specificity. Nevertheless, electrochemical biosensors have some limitations as well (e.g., relatively short lifetime, storage limitations, environmental conditioning, etc.), which have led the researchers into an alternative train of thought. The replacement of the biological recognition element for a synthetic one, such as nanomaterials or molecularly imprinted polymers (MIPs), seeks the same electrocatalysis advantages and high selectivity, avoiding sensitive biological parts. Hence, the electrochemical lactate-sensing field has been greatly expanded during the last years due to the so-called non-enzymatic sensors [35].

For all the above-mentioned discussion, we think that an update of the current approaches towards electrochemical real lactate monitoring should be addressed. In this regard, herein is presented a critical review focused on electrochemical (bio)sensing of lactate in biomedical and agrifood ambits. Alternatively, the readers are kindly referred to others recent reviews, in case their final aim is focused on electrochemical lactate (bio)sensors exclusively applied to medical applications [34,36] or to a more general insight about lactate detection that is not limited to the sensors field [37].

## 2. Electrochemical (Bio)Sensor Transduction Approaches

Table 1 offers an insightful picture of how the state of the art for lactate electrochemical (bio)sensors is currently drawn in the literature. 

As can be appreciated from the date exposed in Table 1, different strategies can be employed in the development of electrochemical lactate (bio)sensors. There are authors employing several electrode materials, such as carbon-based ones, Au and Pt, among others. According to the design of the sensor, a great variety is also observed. The usual approach is to employ enzymes to supply the required selectivity, but some other researchers have successfully obtained enzyme-free lactate sensors, using alternate approaches (nanomaterials, metallic organic framework, molecularly imprinted polymers, etc.). It is also very common to employ membranes to avoid interferents in the target samples; chitosan and Nafion are frequently employed for this purpose. Additionally, similarly to other current research on sensors, nanomaterials (e.g., carbon nanotubes and metal nanoparticles) also play a major role in advanced sensor architectures aiming at enhanced figures of merit.

However, it is possible to agglutinate a great number of sensor architectures in three great blocks depending on their transduction mechanism, namely amperometric, potentiometric and conductometric (Figure 2). These transduction strategies are discussed below.

### 2.1. Amperometric Lactate Biosensors

Among the most commonly employed sensors in lactate detection, amperometric (bio)sensors monitor the current between a working electrode (WE) and a counter electrode (CE); meanwhile a potential is being applied between WE and a reference electrode (RE) (Figure 2a).

Recorded current can be correlated with the bulk concentration of the electroactive species or its production/consumption in the studied sample. The reader may revise the deep and instructive review from Nikolaus et al. to find more fundamental information [62]. Amperometric sensors are considered as a direct and simple tool, where the main bottleneck is the design of a suitable WE. In these cases, it is necessary to evaluate several factors such as the electrode material and its consequently chemical modification, among others. However, amperometric lactate biosensors are not so straightforward in their fundamentals. Enzymes are frequently bonded to the electrode material to enhance the specificity of the resulting sensor [62]. In order to take advantage of the lock and key enzymatic mechanism, new electroactive species are assessed instead of lactate itself. For example, consider the two most employed enzymes for this purpose [5], namely L-lactate oxidase (LOx) and L-lactate dehydrogenase (LDH), and their catalytic reactions exposed below:(1)$${L-\mathrm{lactate}+O}_{2}\overset{L-\mathrm{Lactate}\;\mathrm{Oxidase}}{\to }{\;\mathrm{pyruvate}+H}_{2}O_{2}$$
(2)$${L-\mathrm{lactate}+\mathrm{NAD}}^{+}\overset{L-\mathrm{Lactate}\;\mathrm{dehydrogenase}}{\to }\;\mathrm{pyruvate}+\mathrm{NADH}$$

Both scenarios involve the transformation from L-lactate to pyruvate, using either oxygen or the corresponding cofactor (NAD^+^), and the production of a secondary electroactive specie during the enzymatic reaction, H_2_O_2_ and NADH (reduced form of nicotinamide dinucleotide) for L-LOx and LDH, respectively. Importantly, both species’ concentrations will be easily correlated with the initial lactate in the biosensor surroundings, and more importantly, their oxidation will provide a current that is only attributable to the target analyte. The trained reader may notice that the biosensor will inherit a certain dependency from the enzymatic choice. Oxygen or NAD^+^ must be available in the medium for the correct biosensor performance. This is not a negligible detail, and it should be properly studied. Nevertheless, this ideal free of interferent current also possesses a considerable flaw. The potential to re-oxidise both species is relatively high (ca. ≥ 0.6 V vs. Ag/AgCl reference electrode), even with suitable electrode materials (e.g., Au and Pt), and it may affect other common substances in the matrix studied (e.g., ascorbic acid, uric acid, dopamine, etc.). 

Researchers have found many alternatives to minimise this issue. Simple but effective approaches rely on a Nafion layer onto the surface of the electrode that rejects anionic compounds prone to be oxidised in the WE. On top of that, a polymeric layer (e.g., polyurethane, chitosan, etc.) can be also drop-casted to hinder the diffusions of susceptible substances. Burmeister et al. claimed an expansion in the linear range of the resulting sensor and an interferent-free signal by using this concept [63]. In a more recent work, Lee et al. presented an amperometric biosensor based on the immobilisation of LOx through polypyrrole and polydopamine layers [38]. According to the authors, the preparation was performed in a one-pot pathway, and it was also possible to immobilise glucose oxidase by using this approach in a similar electrode. Lee et al. claimed that the employment of both polypyrrole and polydopamine greatly enhanced the sensitivity of the resulting device. Similar to other studies, a permselective layer made of polyphenol was placed onto the surface of the modified electrode to improve the selectivity of the sensor. Concerning the results obtained, a sensitivity of 3.30 µA mM^−1^ cm^−2^ and a linear range of 0–0.5 mM were found. Indeed, it seems that this sensitivity and stability are acceptable. Furthermore, the permselective layer dwarfs interferent issues, even using an oxidation potential of 0.7 V vs. Ag/AgCl. Nonetheless, its short linear range limits the direct and undiluted application, although the authors should be praised for developing such as fast and simple manufacturing methodology. Booth et al. considered growing platinum black in carbon-based conductive homemade fibres to build their amperometric biosensor [39]. The authors benefited from the catalytic effect of the deposited Pt towards H_2_O_2_. In addition, a layer of poly-m-phenylene diamine was also electrodeposited. Finally, a hydrogel of poly(ethylene glycol) diglycidyl ether loaded with LOx was integrated in the sensor by dip coating. These polymers act as a protective layer towards interferents such as serotonin, ascorbic acid and dopamine, among others. It is noteworthy to mention that a similar fibre was modified with iridium oxide (IrOx) to serve as a pH probe. Booth et al. stated that the sensor was able to measure concentration in the physiologically relevant range and a limit of detection of 19 µM was obtained from the resulting device. Remarkably, the amperometric biosensor was used to evaluate damaged tissue in mice brains, and local lactate changes were recognised and monitored. Thus, the authors proposed this device as possible implantable biosensors.

Other authors have also explored the electrocatalytic effect of nanomaterials. Istrate et al. employed a ternary composite based on gold nanoparticles, reduced graphene oxide and polyallylamine to immobilise LDH onto a screen-printed carbon electrode [41]. Certainly, gold nanoparticles have been stated in other works as an excellent material to build lactate biosensors, which is even more convenient in tandem with carbon nanomaterials [64]. Istrate et al. reported a resulting device with an extensive linear range (4–16 mM), a low limit of detection (1 µM) and high sensitivity (0.28 µA mM^−1^ cm^−2^). With the aim of food application, several specific interferents were assayed, such as acetic acid, ethanol, glucose, ascorbic acid and glutamic acid. Only ascorbic acid was considered an interferent, and only while it was in the same concentration range than lactate. Finally, the sensor was applied to real samples of yoghurt and wine, obtaining excellent outputs. However, it should be noticed that reported operational range cannot be considered as a real linear range. An issue of enzyme saturation is likely to be found in this case. Moreover, the error bars found in the calibration measurements obstruct an accurate real value obtention. Nonetheless, the convenience of the device in applications of foodstuffs such as dairy and wine products can be still considered.

In all previous scenarios, the oxidation potential employed to carry out the measurement is high due to the H_2_O_2_ oxidation requirements (even with the electrocatalytic effect of Pt or similar electrode materials). Current trends are based in diminishing the potential required to perform the chemical reaction. In this regard, redox mediators have left a relevant footprint in the field of lactate detection, and a wide range of possibilities can be found around this topic. Concerning which feasible redox mediator to employ, it is possible to classify them according to their nature: (i) conducting polymer-based mediators (e.g., poly(aniline)–poly(vinyl sulphonate) [65]), (ii) organic dye-based mediators (e.g., Meldola blue [66], tetrathiafulvalene [67]) and (iii) transition metal compound-based mediators. Within this last group, Prussian blue (PB) stands out as the most extensively applied in amperometric lactate biosensors [68,69]. Remarkably, PB’s electrocatalytic property towards H_2_O_2_ oxidation is three orders of magnitude higher than that of regular Pt materials [70]. Thus, it makes the biosensor performance possible by applying a near-zero potential. This is translated in a narrow working potential window where interferences are greatly avoided. Recently, PB has been successfully implemented in devices with a high technological readiness level (TRL). In this sense, Gao et al. presented fully integrated wearable sensor arrays for sweat metabolite monitoring [44]. Glucose, lactate, Na^+^ and K^+^ were evaluated in the multianalyte platform. Regarding the lactate sensor, a thicker PB layer was electrodeposited onto a Au electrode. Interestingly, the authors indicated that thicker PB layers contribute to expand the resulting linear response range of the sensor. In addition, LOx was placed between 2 layers composed of chitosan (permeable film) and carbon nanotubes. This combination produced a very extensive linear response range (5–30 mM) able to cover the physiological concentration in sweat, even during intense exercise. Furthermore, a negligible interference of other species (e.g., ascorbic acid, uric acid, etc.) was appreciated in the sensor, according to the authors. Importantly, the multiplatform was successfully applied in real-time physiological conditions in human subjects at different exercise intensities. The results obtained demonstrated the suitability of the developed sensors. Particularly, lactate amperometric sensor supplied coherent values in all cases, but it was noted that lactate concentration also depended on the lactate excretion and the sweat rate. In the same vein, Vinoth et al. reported a fully printed microfluidic wearable device to monitor lactate, pH, Na^+^ and K^+^ [40]. In this regard, a multianalyte platform is also presented to assess sweat ions and metabolites. Concerning which lactate amperometric biosensor PB nanoparticles were employed as redox mediator layer, the authors remarked that PB nanoparticles possessed a higher surface volume ratio, which entail improvement in the catalytic and sensing performance. Additionally, single-walled carbon nanotubes were deposited on the surface of the electrode to promote the catalytic effect and increase the surface. Similar to other authors, Vinoth et al. also used chitosan as a permeable layer [40]. However, in this case, chitosan was used as immobilisation matrix for PB nanoparticles as well as for LOx. Furthermore, an additional chitosan layer was placed on top of that to prevent enzyme/mediator leaching. A limit of detection (0.2 mM) and remarkable linear response range (1–25 mM) were obtained. In this case, the potential applied during chronoamperometry was −0.17 V vs. Ag/AgCl pseudoreference electrode. Furthermore, the resulting sensor was also evaluated in microfluidic channels. Sweat samples analysed by this methodology offered coherent values in all cases. Temperature and pH influence were also assessed, revealing an improvement margin by using a temperature and pH correction. It should be stressed that reliable measurements in real scenarios (e.g., factories, home, etc.) may require the assessment of environmental factors, especially temperature. This kind of correction approach steps closer to the final application of the device, and therefore, authors should be praised for that. On the other hand, interferents studied (uric and ascorbic acid and glucose) did not affect the lactate determination. Only a slight drift was noted during successive calibrations. According to the authors, this could be attributed to an active material leaching, an issue that may be solved by using a crosslinking methodology to immobilise the enzyme.

### 2.2. Potentiometric and Conductometric Lactate Biosensors

Although less common, it is also possible to determine lactate through potential monitoring instead of using an intensity current. Potentiometric sensors measure electrical potential difference between two different electrodes (WE and RE) both embedded in the same media (Figure 2b). In this case, WE is usually based on a modified electrode with a certain ion-selective membrane (ISM), generating the so-called indicator electrode or ion-selective electrode (ISE). ISEs allow us to correlate the activity of a certain species with the potential recorded, according to the Nernst equation. Particularly, this measurement is carried out under near-zero current. In this scenario, the transduction mechanism is based on the charge carriers free to move inside the ISM (target ions) and its conversion to electrons in the interface of the electrode. This process requires assistance of the RE acting as a contact. Classically, a Ag/AgCl wire immersed in a solution with a known amount of Cl^−^ can be used as a liquid contact. Nonetheless, current potentiometry is ruled by all-solid-contact electrodes (e.g., conducting polymers or carbon nanotubes) due to their undeniable advantages such as lower detection limits and miniaturisation possibilities, among others [71]. The first attempt to apply potentiometry to lactate determination was carried out in 1979 by Shinbo et al. The authors immobilised *LDH* in a gelatine to catalyse the oxidation of lactate by ferricyanide; hence, the variation in the ferricyanide/ferrocyanide concentration ratio could be used to monitor lactate concentration with the resulting potentiometric biosensor [45]. More recently, Lupu et al. developed a more advanced lactate potentiometric sensor, also based on *LDH*. In this work, plasma-enhanced chemical vapour deposition and colloidal lithography were used to deposit nanostructured polyacrylic acid (PAA) onto the surface of the WE and build a suitable immobilisation environment. It is widely accepted that the immobilisation of an enzyme in a nanostructured material provides several advantages, mainly the possibility of a superior enzyme loading due to the nanomaterial surface increase. In this case, the resulting potentiometric sensor provided acceptable sensitivity (49.7 mV per decade) and limit of detection (2 × 10^−7^ M). However, the linear range obtained is very limited for real applications without considering diluting factors [47]. The authors pointed out possible applications to the brain environment where the lactate concentration can reach the nanomolar range, but other considerations, such as sampling methodology and possible interference species in that media, among others, must be addressed first. Interestingly, Ibupoto et al. developed a potentiometric sensor based on ZnO nanorods as immobilisation matrix for LOx. Interestingly, ZnO materials expose strong bonding with enzyme/ionophore membranes, improving the efficiency of the catalytic effect as well as enhancing the flow of the testing substance through the sensor; thus, the output signal is meaningfully magnified. This, altogether with the superior surface of ZnO nanorods, would stand as a suitable environment for immobilisation [47]. The resulting potentiometric sensor possessed adequate sensitivity (41.3 mV per decade) and a linear range between 1 × 10^−4^ and 1 mM of lactate. Despite these average results, the authors reported a superior lifetime (more than 3 weeks) in terms of enzymatic biosensors and good selectivity towards several common interferent species (e.g., ascorbic acid, urea and glucose). 

Even less common, but still viable, is the possibility to determine lactate through a field-effect transistor device, where the sensing element—usually a lactate-related enzyme—modifies the gate of the transistor and an outer sensor interrogates the sample (Figure 2c). For example, Minamiki et al. took advantage of this methodology to fabricate an organic field-effect transistor for lactate determination [49]. LOx and an osmium redox polymer composite were deposited as sensing membranes on the gate. The authors reported a negligible effect of usual interferences such as MgCl_2_, CaCl_2_, NaCl, p-cresol, urea and glucose. In addition, a relatively high linear response range was found (1–10 mM) due to the miniaturisation of the extended gate, according to the authors. Despite the fact that the authors examined interference species (e.g., p-cresol, urea, and glucose) in similar sweat concentrations, no real sample was assayed. Furthermore, it should be noticed that the linear range is not enough to cover lactate concentration during intense exercise (ca. 20 mM). Therefore, the device should be reconsidered in another biological fluid or real sample matrix (e.g., foodstuff), otherwise dilution is mandatory. In the same field, Schuck et al. presented a common-gate field by using a graphene-Nafion material to modify the gate of the transistor [48]. In this case, the sensing element was the LDH enzyme mixed with glutaraldehyde as the crosslinking element and covered by a chitosan layer. According to the authors, this configuration was chosen to maximise the binding stability as well as to hinder possible interferent species. Interestingly, the authors made a comparison with LOx and stated that LDH was more convenient for field-effect transistors, since the hydrogen peroxide production of LOx has no partial charges. The resulting device provided a linear response range of 0.25–10 mM and was also evaluated in human fluids such as human plasma and serum. On the one hand, plasma analysis presented higher sensitivity than serum, according to the authors. On the other hand, an acceptable correlation coefficient (R^2^ = 0.9787) was obtained, proving the suitability of the device. In addition, uric and ascorbic acid were evaluated as interferents, as well as glucose, due to its potential doping effect over the active layer, according to the authors. In any case, lactate determination was affected by these compounds, corroborating the suitability of the field-effect transistor as an accurate tool for clinics.

### 2.3. Enzyme Immobilisation Influence and Viable Approaches

The reader may have noticed that there is not a clear and straightforward methodology to immobilise the enzymes on the biosensors. The reality shows that the research community is still exploring the possibilities of this field because a clear improvement still exists in it. In fact, enzyme immobilisation should not be never considered as a meaningless factor. Long-term stability and sensitivity are definitely related with the immobilisation methodology. In addition, enzyme leaching and the decrease in enzyme activity by deactivation can be diminished using the proper approach. In the literature, immobilisation methods can be classified as follows: (i) physical adsorption, (ii) physical entrapment, (iii) covalent binding and (iv) crosslinking-based methods [34]. 

Each one of them possesses intrinsic pros and cons. For example, physical adsorption of the enzyme (Figure 3a) is the simplest methodology. It can be carried out by using the drop-casting method to dry a drop of enzymatic mixture or employing the dip-coating approach to put the enzyme and the electrode substrate (modified or not) in contact for a certain period of time. Only van der Waals forces are involved in the attachment of the enzyme. On the bright side, enzymes are not altered, so their enzymatic activity is maintained. However, it is possible to deduce the evident flaw of this approach: the retaining capacity. Interaction between the enzyme and the solution results in a slow but relentless dissociation of the enzyme from the modified electrode, leading to a decrease in the enzymatic activity, and hence to the sensitivity of the biosensor. Thus, physical adsorption is not frequently currently employed [72,73]. A viable option is to improve the retainment consist of the entrapment of the enzyme using a physical barrier (Figure 3b). Enzymes can be immobilised in a polymer or sol–gel placed on top of the substrate. Controlled polymerisation can be achieved by using electropolymerisation or photo-binding, among others. Contrarily to physical adsorption, the enzyme is perfectly confined in the matrix, which provides minimum leaching as well as excellent additional features such as mechanical, thermal and chemical stability. Nevertheless, the drawback resides here on how the membrane affects the diffusion of the analyte through the biosensor, resulting in a decrease in the sensitivity. Recently, Lee et al. employed this methodology to immobilise LOx in a polypyrrole/polydopamine polymer matrix, entrapping the enzyme within it [38]. It is also possible to enhance the retainment of the enzyme via chemical bonding. Covalent binding (Figure 3c,d) methodology consists of the creation of covalent bonding between the functional group and the substrate. Usually, amino acid residues are used for this purpose, and some reverse inhibitors can be employed as well in order to protect the enzyme active centre. In theory, this bond can be made in any substrate prior to its modification with functional groups [74]. The resulting enzyme–substrate union can be oriented in a specific manner (Figure 3c) or randomly (Figure 3d), but in both scenarios, a stable monolayer is formed. As an illustrative example, Lupu et al. described the covalent binding between LDH and a polyacrylic acid layer [47]. This layer was previously activated by using N-hydroxysuccinimide and 1-ethyl-(3-dimetylaminopropyl)carbodiimide to generate (COOH-) free spots where the covalent union can be performed. Nonetheless, the sensitivity of the sensor will depend on the enzyme loading, which in this case is related with the available surface to bind the enzyme. Enhancement of the surface is pursued in order to increase the number of enzymes bonded. The main option relies on the modification of the surface with nanomaterials, especially carbon nanomaterials (e.g., carbon nanotubes, graphene, etc). Otherwise, the increase in enzyme loading can be performed by constructing a three-dimensional grid using a bifunctional agent, which agglomerates a great number of enzymes. This process is known as the crosslinking method (Figure 3e). Due to the intense binding, an enhanced stability and higher enzyme loading are obtained. The resulting 3D grid depends on the bifunctional group (e.g., glutaraldehyde) and the initial protein used. Chan et al. described a crosslinking methodology using a combination of bovine serum albumin (BSA) and glutaraldehyde (GA) to immobilise simultaneously lactate dehydrogenase and pyruvate oxidase onto a carbon electrode [2]. Importantly, the sensitivity of the biosensor based on crosslinking may not be as high as expected. Even though the number of enzymes is higher, it should be noticed that the multiple unions of the enzyme with the bifunctional group can lead to an alteration of its structure, decreasing the enzymatic activity.

### 2.4. Non-Enzymatic Sensors

The main reason to use enzymes in lactate biosensors is to provide a high specificity towards this species. However, as the core of the lactate biosensor, the biological recognition element possesses a critical role in its performing. Thus, the above-mentioned possibilities should be carefully assessed in order to obtain the most robust and sensitive device in each scenario. Moreover, enzymes will be always affected by other parameters such as pH and temperature, which may not be adequate in other applications. It is true that for medical applications, pH and temperature remain in physiological conditions, ensuring enzymatic stability. Nevertheless, this is not so convenient in sport medicine, where sweat pH can change dramatically; and is much more problematic in the agrifood field, which possesses a wider range of pH and temperature. This influence can be decreased by using membranes to make contact between the enzyme and the surroundings more difficult [75]. On the other hand, other authors have decided to explore an alternative path and have tried to propose a lactate sensor without this biological recognition element in order to avoid these dependences. The resulting devices are the so-called non-enzymatic sensors, sensors which employ other materials to provide selectivity to the system. In general terms, three different currents can be appreciated based on the materials used: (i) nanomaterials, (ii) metal–organic frameworks (MOFs) and (iii) molecularly imprinted polymer (MIP)-based sensors (Figure 4).

In the first place, specific features of nanomaterials are exploited to promote lactate oxidation mechanism over other species. For example, Change et al. proposed a non-enzymatic lactate sensor by using cobalt oxide nanostructures [51]. The authors indicated that the material possessed Co^2+^ and Co^3+^ centres while the material was embedded in solution. The presence of both centres oxidised lactic acid into pyruvic acid in a selective manner. They also pointed out that the electrochemical signal was directly related with the morphology of the catalyst. Concerning to the results obtained, a linear response range from 0.01 to 3 mM was reported, as well as a limit of detection of 6 µM. Regarding the influence of interferents, several species at relatively high concentrations were evaluated during lactate calibrations, such as citric acid, glucose, maleic acid, tartaric acid and urea. According to the authors, no significant interferences were detected. In the same vein, Kim et al. presented a porous nanostructured nickel oxide-based sensor to promote selective lactate oxidation [50]. In this case, the effect of the calcination temperature and the subsequent nanostructure generated was also studied. In this regard, a temperature of 250 °C was chosen as the optimum value to obtain the selective nanomaterials to integrate into the sensor. The device provided sensitivity of 62.35 μA mM^−1^ cm^−2^, a limit of detection of 27 μM and a linear response range of 0.01–7.75 mM. Interestingly, the calibration was carried out at a potential of +0.55 V. However, no influence of other interferent species (ascorbic acid, uric acid and dopamine) was noted. It is also possible to find in the literature examples of composite nanomaterials used to design the sensor architecture. Hussain et al. proposed a nanocomposite based on CuO and multiwalled carbon nanotubes with a Nafion layer [52]. The authors proposed a lactate oxidation mechanism similar to the one found in LOx. Briefly, lactate was oxidised to pyruvate in the presence of oxygen, and also produced hydrogen peroxide. This hydrogen peroxide is decomposed into oxygen in a two-step mechanism where two electrons are released. Hence, the current recorded is attributable to this electron flow. A relatively extensive linear range response (100 pM–10 mM) and an excellent limit of detection of 88.5 pM were reported. Moreover, lactic acid determination was satisfactorily achieved in the presence of other biological molecules such as cholesterol, dopamine or testosterone, among others. In contrast to other works, Hussain et al. used a current voltage monitoring instead of a classic amperometry. As can be appreciated, non-enzymatic lactate sensors based on nanomaterials can be considered as interesting and innovative alternatives in lactate sensing. Nonetheless, if the reader carefully examines the works in this pipeline, they could notice an important fact. In the majority of these works, lactate sensors should work in an alkaline media to promote the selective lactate oxidation, which is usually guaranteed due to a concentrated NaOH media. Therefore, there is still a pH sensor dependence similar to the one found in enzymatic sensors, which must not be overlooked. 

Concerning the lactate sensors based on MOFs, these structures are also located in the nanomaterial kingdom, and they can be understood as a crystalline material built through the coordination of metal ions/clusters with organic linkers [76]. The trained reader may assume that there is an unusual freedom to construct the desired MOF by using different building blocks. In fact, this is the core of the application of these materials in sensing, their tuneable properties. Regarding their application in lactate determination, Wang et al. reported a 2D-oriented MOF of Cu_3_(btc) nanocubes modified with freestanding amino-functionalised graphene nanocomposite to prepare a lactate and glucose sensor [57]. According to the authors, the excellent features of the resulting sensor allowed for the simultaneous detection of both analytes by using cyclic voltammetry. Contrarily, for the determination of each species, a specific potential was selected for the amperometry assays, namely −0.10 and 0.65 V for lactate and glucose, respectively. A wide list of interferent species either ionic or organic molecules was evaluated, but no concerning effects were reported. For lactate sensing, a linear dynamic range of 0.05–22.6 mM and a limit of detection of 5 μM were found. In addition, enhanced lifetime stability of 50 days was also indicated. Finally, the sensor was applied to real sweat samples, obtaining acceptable outputs. Interestingly, the sensor also possessed excellent performance under flexibility tests, proving its feasible future applications in the wearable sensing field. However, it is noteworthy to mention that the application of this device to another field, such as clinical or agrifood environments, should involve a reassessment of the interferents. The reported studies employed a concentration of organic species of 1 μM, which is a relatively low concentration value for a real sample matrix in both ambits. The consequent results will state if this application is feasible or not.

The last alternative approach to avoid enzymes is based on MIPs. In brief, these materials are produced via polymerisation by using the target analyte as a template for the resulting polymer; later, the template is removed, leaving cavities with the exact shape to rebind a target analyte molecule, in a similar manner to how enzymes work (lock-and-key mechanism) [77,78]. In addition to the evident selectivity, molecularly imprinted polymers also bestow the sensor with high reliability and either mechanical or chemical stability. Despite these advantages, there are not many studies related with sensors based on MIPs for lactate-sensing applications. In this sense, the work developed by Pereira et al. should be praised. In this case, electropolymerisation was used to fabricate the MIP, granting better control during the polymerisation and better adhesion to the electrode material [59]. Prior to the polymerisation, the electrode material was modified with reduced graphene oxide and gold nanoparticles to improve the electrochemical performance of the sensor. Before the application of the developed sensor, an electrochemical and a morphological characterisation were performed. The electrochemical characterisation comprised classical studies in cyclic voltammetry as well as electrochemical impedance spectroscopy assays, obtaining in both cases the ideal behaviour for an electrochemical sensor. Moreover, the results obtained by atomic force microscopy revealed a roughness increase in the material when the lactate was inside the structure, but it was later demonstrated that this variation was reversible. Concerning the analytical performance of the sensor, differential pulse voltammetry was employed to determine lactate. A split linear range was reported; the first one comprised lactate concentration between 1 × 10^−10^ and 1 × 10^−9^ M, and the second one from 2 × 10^−9^ and 1.5 × 10^−8^ M. Unfortunately, no clear explanation for this fact was supplied by the authors. On the other hand, a useful rebinding study was provided, exposing the suitable stability of the sensor. Furthermore, the selectivity of the sensor was also evaluated by using diverse organic molecules (lactic acid, acetic acid and glucose, among others), but no relevant response was obtained towards these molecules comparing the molecularly imprinted polymer with a nonimprinted one. Finally, the resulting device was applied to enriched sugarcane vinasse samples. In all scenarios, near 100% recoveries were obtained, revealing the good accuracy of the sensor. Nonetheless, the “dual” and low linear response range and limit of detection (8.9 × 10^−11^ M) values obtained limit the application of the sensor to real samples without considering dilution factors. It is true that the outputs of selectivity are very promising, but they are still far from being considered a strong alternative to classic enzyme-based sensors. Despite this, taking into account the youth of this approach, we can expect to contemplate a considerable change in this trend in the near future.

## 3. Implementation of Lactate Biosensors on Biomedicine: Real-Time Health Care

Generally, the use of lactate biosensors in medicine is aimed at the monitorisation of alterations in metabolism that cause an increase in lactate levels, such as clinical diseases or high-intensity physical activities [1,6,79]. The concentration of lactate in different body fluids is a warning that can be employed in the diagnosis, prognosis and treatment of multiple pathologies. For this reason, research on lactate monitoring in the biomedicine field has been focused during the last few years at the development of tailored simple wearable lactate biosensors for the real-time monitoring of patients in clinical situations. Additionally, multiple wearable devices have been developed for lactate biosensor monitoring during physical athlete training. Currently, some commercial lactate biosensors can be purchased for this purpose (e.g., *Lactate Pro 2* or *Lactate Scout 4*), which have several drawbacks. Mainly, the lactate determination is carried out using blood samples from the patient, which implies a very invasive approach. Moreover, the analysis is noncontinuous, so several samplings must be performed to secure a wide monitoring of the exercise progress. Contrary to this, the new wearable devices recently developed are based on sweat sampling. This biofluid is produced and released uninterruptedly from the skin, particularly during physical efforts, so the performing of the sensor involves a noninvasive methodology and a real-time determination. In addition to this, some of these devices have wireless systems that allow for a fast compilation and treatment of the acquired data in smartphones and other portable tools. The most recent advances in lactate biosensor development reported in the literature are summarised in Table 2. In general, the development of lactate biosensors is restricted by some requirements and limitations, such as quality analytical parameters, interferences or biocompatibility. Multiple approaches have been tested to overcome these specifications. Thus, some of the most interesting advances reported recently are discussed in this review.

One of the first examples of wearable lactate biosensors for physical activity monitoring was reported by Jia et al. in 2013. Interestingly, they assembled a flexible screen-printed LOx-based biosensor that was used as a temporary-transfer tattoo. Briefly, a Papilio temporary-transfer tattoo-based paper was printed with conductive carbon ink for the working and counter electrodes, and with silver/silver chloride ink for the pseudoreference electrodes. Then, the working electrode was modified with tetrathiafulvalene, as redox mediator, and multiwalled carbon nanotubes, followed by the LOx enzymes and a chitosan layer. The lactate tattoo sensor exhibits a highly linear response throughout the 1 to 20 mM range, with high sensitivity (644.2 nA mM^−1^). Moreover, this device showed high operational stability over an 8 h period with highly reproducible results (RSD = 3.60%). Subsequently, the developed device was applied to sweat monitoring. It was reported that the device can endure repetitive mechanical deformations experienced by the epidermis during exercise. In comparison with enzyme-less control sensor, the modified device displayed facile biocatalytic ability toward lactate oxidation in the perspiration and an extremely low noise level [80]. Similarly, Xuan et al. developed a wearable device that includes pH and temperature sensors as well as the lactate biosensor. This on-body device is based on the use of PB as redox mediator, and a polymeric membrane of tetradodecylammonium tetrakis(4-chlorophenyl)borate, polyvinyl chloride and bis(2-ethylhexyl)sebacate, which modulates the lactate flux reaching the enzyme in order to expand the linear response range of the sensor, avoiding an early enzyme saturation. Additionally, it reduced the influence of pH or temperature of the perspiration in the resulting device [75]. The scheme of a wireless wearable lactate amperometric biosensor applied to real-time lactate monitoring on sweat is represented on Figure 5, as well as several examples of wearable sensors with different configurations.

Equally interesting is the wearable platform developed by Liu et al., which can be used to monitor both lactate and glucose simultaneously. To achieve this, a Au electrode upon silk was modified with a graphene and Pt nanoparticle nanocomposite. Then, glucose oxidase and LOx were cast upon this layer in different electrodes and covered with silk nanofibrils and glutaraldehyde to immobilise the enzymes. These features ensure the stability of both sensors, and as result, continuous monitoring stability of lactate up to 23.6 h was achieved. Despite the high sensitivity (6.68 µA cm^−2^ [log_10_ (mM)]^−1^) displayed by the sensor device, the top-limit concentration from the linear relationship was only 6 mM, whereas high concentrations of lactate up to 25 mM were reported in sweat samples [81]. Hence, further research must be carried out to improve the dynamic linear range of this wearable biosensor [82]. Alternatively, a wearable biosensor was reported with the aim of monitoring in another biofluid instead of sweat: a noninvasive mouthguard biosensor for lactate monitoring in saliva. Briefly, a PB-graphite ink was used to print the working and auxiliary electrodes of the biosensor, whereas a Ag/AgCl conductive ink was employed for the reference electrode, carrying out a drying stage at 80 °C for 20 min in both cases. Then, the LOx was immobilised on the working electrode surface by electropolymeric entrapment in a poly(o-phenylenediamine) film. This device displays both low-potential signal transduction and the rejection of coexisting electroactive and protein constituents in saliva samples, which lead to highly sensitive, selective and stable response. In this sense, a negligible effect upon the lactate response (≈5%) was reported for this biosensor in the presence of physiological concentrations of ascorbic acid and uric acid. Moreover, this device exhibits high sensitivity (0.553 mA·mM^−1^) in a linear range of 0.1 and 1 mM, which is competent for the lactate level reported in saliva samples (up to 1.6 mM in extreme-physical-activity conditions) [83]. Finally, the stability of the biosensor device in an untreated saliva sample was assessed, with only small variations of the current signal (90–106% of the original response) observed during a 2 h period [84]. On the other hand, non-enzymatic wearable sensors have been reported as well. For example, a molecularly imprinted polymer was electropolymerised over a carbon working electrode coated with Au nanowires using lactate as imprinted template. Consequently, a sensor with high performance and specificity towards lactate was prepared. The changes in the lactate response of the device caused by several compounds found in sweat were tested. Urea, pyruvic acid and uric acid in physiological concentrations gave only a 1.4% change of the specific response for lactate, whereas the change was negligible for the rest of the studied compounds. This device exhibits a wide linear range (from 1 μM up to 100 mM) and a low limit detection (0.22 µM) as well, but its sensitivity is less competent than enzymatic lactate biosensors. Moreover, this sensor displays extensive storage stability over 7 months at room temperature, overcoming the usual degradation problems of enzymes [60]. Alternatively, Wang and coworkers developed another non-enzymatic wearable lactate sensor by modifying screen-printed carbon electrodes with transition metal-layered double hydroxides. A Co-Ni metal framework was used as a self-sacrificial template to fabricate this hierarchically structural material with uniform porosity and high electrochemically active surface area. Therefore, an outstanding electrocatalytic performance for lactate sensing was achieved with high sensitivity of 4.7 µA mM^−1^. Moreover, for the lactate determination an applied potential of 0.55 V (vs. Ag/AgCl) was used without interferent problems from compounds common in sweat, such as glucose or ascorbic acid. The linear response range (2–26 mM) of the sensor was suitable for the direct analysis of sweat. Hence, samples taken during aerobic and anaerobic exercises were analysed, and the lactate values obtained (12.39 and 23.23 mM, respectively) were consistent with the literature. As expected from an enzyme-free sensor, this device exhibits ultrastable performance, with retention of 98.72% after 28 days [58]. Several examples of wearable lactate sensors with different configurations are shown in Figure 6.

The physiological levels of lactate in the biological fluids are a key parameter when a biosensor is developed. The linear response range (LRR) of the device must cover these values to carry over an accurate assessment of the pathological situation of the patient. It should be reminded that dilutions are not possible in wearable sensors due to the intrinsic concept of these devices. In this regard, the use of porous membranes to control the diffusion of the analytes is an approach on the rise to enhance the LRR of biosensors. For example, the use of a polyamide membrane to embed graphene oxide (GO) nanosheets with LOx immobilised on its surface was suitable arrangement for the increase in the DLR of a lactate sweat sensor up to 100 mM. The lactate levels in sweat can reach even 62.2 mM after exercise of the maximum aerobic power [88]. Therefore, the GO-polyamide biosensor is suitable for the accurate monitoring of physical exercise intensity. However, the limit of detection of this device is 1 mM, so it may be not suitable for other medical monitoring applications [89]. Moreover, this kind of membrane also avoids interferences during the determination of lactate by other electroactive species in the studied fluid. Nafion films, a sulphonated fluoropolymer, are a widely used approach for this purpose due to their avoidance of the diffusion of possible interferents towards the electrode surface. Zhang et al. developed a biosensor of LOx coated with Nafion that has an almost negligible current response when a high concentration of uric acid, amino acid or glucose is added into the device [90]. Moreover, Narayanan et al. used Nafion to create a semipermeable membrane barrier in the developed LDH biosensor that further provides efficient screening of the interference species to enable the device to produce accurate responses in the presence of these compounds. On this point, they prove the selectivity of the developed biosensor when high concentrations of glucose, ascorbic acid and uric acid are in the studied fluid [91]. Alternatively, the use of redox mediators to diminish the oxidation potential of NADH close to 0 V is another useful approach to avoid interferences during the application of an LDH biosensor, as commented previously. Teymourian et al. combine the great electrocatalytic activity of Fe_3_O_4_ nanoparticles with the excellent conductivity of multiwalled carbon nanotubes to develop an LDH biosensor with a working potential of 0.0 V vs. Ag/AgCl. The electrocatalytic mechanism is attributed to the iron phosphate redox system: First, Fe_3_O_4_ nanoparticles are directly reduced at the electrode surface into Fe^2+^ ions that subsequently combine with phosphate ions in the buffer solution to produce FePO_4_, which could be responsible for the observed redox behaviour. Moreover, they prove that a negligible response is obtained with this device when high concentrations of several interferents (e.g., glucose, ascorbic acid, uric acid) are spiked. Despite the great effort of authors, this device has a low maximum output range (0.5 mM), so further research must be carried out to improve this feature [92].

Furthermore, one approach on the rise to enhance the features of lactate biosensors is the application of nanoparticles. In this regard, Nesakumar et al. modified a glassy carbon electrode with carbon-embedded ceria nanoparticles to act as a link between the electrode and an upper LDH layer. The nanoparticles interface favoured the fast electron transfer between both, so the device exhibited high sensitivity and a rapid response time under 4 s [93]. A fast response time is required in lactate monitoring to provide an accurate diagnosis. Otherwise, Narwal et al. prepared LDH nanoparticles through a desolvation method with ethanol and subsequent treatment with glutaraldehyde and cysteamine to provide intermolecular crosslinking of enzyme molecules and functionalised with thiol groups, respectively. The immobilisation of the enzyme on an Au electrode was achieved by using thiol groups between nanoparticles and hydroxyl groups with the metal, which provided a biosensor with high storage time of up to 7 months [94]. Importantly, one of the main drawbacks of commercial lactate biosensors is their low lifetime and storability thorough time. Hence, it is important to develop an approach, such as the one previously mentioned, to avoid the lixiviation of enzymes and improve their long-term stability.

Furthermore, other potential biomedical applications of lactate biosensors have been reviewed. In this regard, Narwal et al. proposed their biosensors for the diagnosis of cardiogenic shock. Several serum samples of healthy patients and those with cardiogenic shock were analysed with this device, and they determined that lactate levels ranged from 0.4 to 2.2 mM in samples from healthy people, whereas several higher concentrations between 12 and 30 mM were estimated for the unhealthy ones [94]. On the other hand, lactate biosensors have been also applied to detect lactate as tumoural biomarkers due to their implication on carcinogenesis and immune tumour evasion. Hussain et al. made great efforts in this regard. They developed a lactate biosensor composed of LDH enzymes immobilised on bifunctionalised conducting polymer of polyaminobenzoic acid composite with N,S-doped porous carbon. Therefore, this device was applied to detect lactate in one noncancerous (Vero) and two cancer (MCF-7 and HeLa) cell lines, finding a higher concentration of lactate in the Vero cell line. Moreover, they studied the cytotoxic effect of a lactate transport inhibitor, α-cyano-4-hydroxycinnamate (αCHC), and its applicability on breast cancer. Decreasing values of lactate concentration of 28, 90 and 42% in Vero, MCF-7 and HeLa cells, respectively, were calculated, proving the suitability of αCHC in cancer growth control [95]. Multiple researchers have carried out comparative studies with lactate biosensors using blood and serum samples of healthy people and lactic acidosis patients [91,96,97,98]. The higher concentration of lactate on patients was stated as a good marker for diagnosis, but sampling is required. This makes real-time monitoring impossible, which is essential in some cases, such as neonatal lactic acidosis. Consequently, the future of lactate biosensors for medical applications lies in the development of wearable devices, such as the ones applied for physical exercise monitoring, which allow for accurate wireless monitoring.

Despite these advances, it is worth mentioning some specific outbreak approaches that already exist in literature. Lactate continuous monitoring is pursued, and wearable systems stand as the suitable choice; however, the limited possibilities to power these devices hinder the development of real-life and sport medicine applications. This is why some authors are devoting efforts in the so-called self-powered sensor platforms. It is possible to find out two main trends in this sense: (i) sensors based on biofuel cells, where a parallel chemical reaction supplies enough energy to maintain the regular performance of the developed sensors [99], and (ii) sensors powered by a piezoelectric material activated by personal movement [100]. Concerning the former alternative, Hartel et al.’s [101] efforts should be praised. The authors developed a resettable sensor platform based on a biofluid cell by using a lactate-oxidisible anode and an oxygen-reducible cathode in tandem with a reversible redox mediator (Prussian blue). In this case, a power generation of 13 µW cm^−2^ was measured, twice the amount estimated to power the adjacent biosensor. However, the self-power path is not restricted to biofuel cells, as suggested the studies published by Mao et al. [100]. A piezoelectric material based on polyvinylidene fluoride and tetrapod-shaped ZnO was integrated in the biosensor platform to take advantage of personal movement and power the device. Remarkably, an advance device was successfully tested on a professional athlete to evaluate his maximal lactate steady state. Additionally, the results were contrasted with Lactate Scout, obtaining a promising correlation. Unfortunately, their final device is limited to scenarios where an intense movement is recommended, thus restricting possible clinical applications. 

On the other hand, there is another breaking point in research where lactate sensors are also involved: the concept of the Internet of things (IoT). The revolutionary idea to connect “things” between each other and with the Internet at the same time is becoming a megatrend in many engineering and scientific ambits [102]. In this regard, lactate sensor platforms can be considered as a great asset to include in this philosophy, and many authors are stressing this issue. Lin et al. recently published a work in this sense [103] reporting a lactate- and caffeine-sensing platform employing hydrogel to drive the sweat sample obtained by natural perspiration. This platform possessed a wireless readout circuitry able to upload the performed measurements in a customised data cloud, a convenient approach to integrate the platform later within the IoT concept.
biosensors-12-00919-t002_Table 2Table 2Most relevant cases in the employment of lactate biosensors in biomedicine and sport.BiosensorElectrodeImmobilisation ProcessLRR (μM)Sensi. (µA/mM)LOD (µM)Tr (s)Lifetime (Days)Samp.App.Ref.Lactate Oxidase BiosensorsMWCNT/TTF/LOx/ChitCarbon inkCrosslinking1000–20,0000.644--152SweatPhysical exercise intensity monitoring[80]PB/LOx/Chit/AuNWsAuNWs-0–30,0000.69137106SweatPhysical exercise intensity monitoring[104]LOx/BSA/PEGDE/β-cysteamine/AuNNs/AuAuCrosslinking1000–25,0000.6554-28Sweat-[105]LOx/PtNPs/GO/Au/SFNFsAuEntrapment400–6000----SweatPhysical exercise intensity monitoring[82]ETH 500-PVC- DOS/LOx/PB/SPECarbon ink-1000–25,0000.009411050-SweatPhysical exercise intensity monitoring[75]LOx/PANHS/GO/Pd/PolyamidePdCrosslinking1000–100,000-1000--Sweat-[89]Nafion-LOx/PPy/MWCNT/PA6Pt-0.001–1000--0.8-SweatPhysical exercise intensity monitoring[90]PDDA/LOx/ZnO/MWCNT/PGPGAdsorption200–20007.366120Serum-[106]LOx/sol-gel/MWCNTs/GCEGCESol–gel200–20006.0310.3528Serum-[107]PB-PPD-LOx-mouthguardPrussian blue graphite inkElectropolymeric entrapment100–10000.0005550--SalivaHealth and physical exercise monitoring[84]Pt/o-PD/PEG/BSA/Chit-LOX-Pt-Ceria-AOPtAdsorption0.0001–15,500-0.0001621Rat tissuesMonitoring in vitro and in vivo tissues during hypoxia conditions[108]LOx/cMWCNT/CuNPs/PANI/PGEPGECovalent1–2500-0.255140PlasmaLactate acidosis diagnosis[96]Nafion/LOx-GO-Ch/PB/SPEGraphite-1000–50,0000.0720.02--Buffer solution-[109]Au/MoO_3_/LOx/NafionAu-500–80000.871501016--[110]HRP-PEGDGE-Os/Chit-LOx/polyphenolGraphite pasteCrosslinking100–10000.76313-91Saliva-[111]MWCNT/FcMe/Chit/HRP/BSA/LOx/SPBGESPCEEntrapment30.4–243.93.4222.6-150Embryonic cell cultureGrowth evaluation of embryo[19]Lactate Dehydrogenase BiosensorsLDH/RGO-AuNPs/SPCESPCEEntrapment10–5000770.13825SerumCancer biomarker detection[112]LDHNPs/AuAuCovalent binding0.01–55,00010.830.012.5210SerumCardiogenic shock diagnosis[94]LDH/GrONPs/PGEPGECovalent binding5000–50,000-0.1560SerumLactate acidosis diagnosis[97]AuNP-cysteamine-LDH/Nafion/MWEW-500–70002.45411-18SerumLactate acidosis diagnosis[91]LDH-NAD^+^/Fe_3_O_4_NPs/MWCNTs/GCEGCECovalent binding50–5007.675-14Serum-[92]LDH/MWCNTs/Chit/AuAuCovalent binding0–120-15810BloodLactate acidosis diagnosis[98]LDH/MWCT-MBCPECrosslinking100–10,0000.427.5--BloodPhysical exercise intensity monitoring[113]LDH/MG/SWNT/GCEGCECrosslinking200–10,0000.0256160-8Rat cardiomyocyte cell cultureMonitoring of cardiomyocytes during hypoxia[114]LDH-NAD^+^/pTTABA/DPCDPCCovalent binding0.5–40000.020.112-60Extracellular matrix of cancer cellsCancer diagnosis and antitumour activity evaluation[95]LDH-GPT/SPCESPCE-100–10000.0335300-Cell cultures, sweatGrowth evaluation of cells, physical exercise intensity monitoring[115]NADH/LDH/Nano-CeO_2_/GCEGCEElectrostatic interactions200–2000571.19504-Buffer solution-[93]Other Enzyme and Non-enzyme SensorsFC b_2_/nAu-AuAu-300–20005.33-*-*91Sweat, saliva-[116]MIPs-AgNWsCarbon-1–100,0000.00450.22-212SweatPhysical exercise intensity monitoring[60]Cu_2_(NDC)_2_/PDHPPDHP-50–22,250114255-Sweat-[117]NH_2_-GP-Cu_3_(btc)_2_GP-0.05–22.6-5--Sweat-[57]SPCE-NiCo (layered double hydroxide)SPCE-2–264.70400-28SweatPhysical exercise intensity monitoring[58]Commercial Biosensors (Lactate Oxidase-Based)StatStrip^®^ Lactate--300–20,000--1391 Blood (0.6 µL)Medical monitoring[118]StatStrip Xpress^®^ Lactate--300–20,000--1391 Blood (0.6 µL)Medical monitoring[118]Lactate Plus Version 2--300–25,000--13-Blood (0.6 µL)Physical performance monitoring[119]LactatEDGE--700–22,000--4591 Blood (0.3 µL)Physical performance monitoring[120]Lactate Pro 2 (LT-1730)--500–25,000--15-Blood (0.3 µL)Physical performance monitoring[121]Lactate Scout 4--500–25,000--10
Blood (0.2 µL)Physical performance monitoring[122]Biosen C-Line (glucose and lactate)--500–40,000--20–4550Blood, plasma or serum (20 µL)Medical monitoring[123]AO: ascorbate oxidase; App.: application; AuNPs: gold nanoparticles; AuNWs: gold nanowires; BSA: bovine serum albumin; Chit: chitosan; cMWCNT: carboxylated multiwalled carbon nanotubes; CPE: carbon paste electrode; CuNPs: copper nanoparticles; DOS: bis(2-ethylhexyl) sebacate; DPC: doped porous carbon; ETH500: tetrakis(4-chlorophenyl) borate; FC b_2_: L-lactate:cytochrome c oxidoreductase; FcMe: ferrocene methanol; GCE: glassy carbon electrode; GO: graphene oxide; GP: graphene paper; GPT: glutamate pyruvate transaminase; GrONPs: graphene oxide nanoparticles; HRP: horseradish peroxidase; LRR: linear response range; LOD: limit of detection; MB: Meldola blue; MG: methylene green; MWCNT: multiwalled carbon nanotubes; MWE: microwire tungsten electrode; NAD^+^: adenine dinucleotide; nAu: gold nanoclusters; o-PD: o-phenylenediamine; PA6: nylon; PANHS: 1-pyrenebutyric acid–N-hydroxysuccinimide ester; PANI: polyaniline; PB: Prussian blue; PDDA: polydiallyldimethylammonium chloride; PDHP: pencil drawing hydrophobic paper; PEG: polyethylene glycol; PEGDE: poly(ethylene glycol)diglycidyl ether; PEGDGE: poly(ethylene glycol)diglycidyl ether; PG: pyrolytic graphite; PGE: pencil graphite electrode; PGE: pencil graphite electrode; PPy: polypirrol; PtNPs: platinum nanoparticles; pTTABA: (poly 3-(((2,2′:5′,2′’-terthiophen)-3′-yl)-5-aminobenzoic acid; pTTCA: poly-5,2′-5′,2-terthiophene-3′-carboxylic acid; PVC: polyvinyl chloride; Samp.; samples; Sensi: sensitivity; RGO: reduced graphene oxide; SFNFs: silk fibroin nanofibrils; SPBGE: screen-printed graphite electrodes; SPE: screen-printed electrode; SWNTs: single-walled carbon nanotubes; Tr: response time: TTF: tetrathiafulvalene.


## 4. Rise of Lactate Biosensors on the Food Industries: Proficient Quality Control

Recently, several potential applications of lactate biosensors in different industries of food production have arisen. Lactate level is a significant parameter in the evaluation of the stability, quality and organoleptic characteristics of different foods. In this regard, Figure 7 shows a schematic summary of the most interesting implementation of these devices in the agrifood field. The role of lactate in the wine industry is of major note. Malic acid is converted to lactate during malolactic fermentation, a crucial step of the vinification that leads to deacidification and softening of the wine taste. Moreover, the quantity and proportion of this compound can be used to avoid adulterated products [124]. In fact, quality control of beer and cider is carried out via lactate monitoring as well [125,126]. Fermentation and bacteria contamination is studied to avoid excessive acidification and the occurrence of unpleasant tastes. Similar inspections are performed in dairy and juice production to prevent the contamination of lactic acid bacteria. The monitoring of lactate with biosensors will allow for the detection of decayed products or critical points during manufacturing, where secondary contamination takes place, supplying meaningful insights of the process and the final products [127,128].

Remarkably, lactate assessment is carried out in the meat industry as well. During postmortem anaerobic glycolysis, lactate and several other compounds are produced. These compounds determine the quality of the product (e.g., sensorial properties, antioxidant capacity and wateriness). Hence, the use of lactate biosensors has been proposed to evaluate the organoleptic properties of meat products [129,130,131]. Equally interesting is the application of lactate biosensors in aquaculture. Blood lactate in fish can be monitored to evaluate stress conditions, which could have a negative impact on the security and quality of final products [132].

Some of the most recent and relevant advances in the application of lactate biosensors in food samples reported in the literature are summarised in Table 3. One of these proposed applications is the control of previously mentioned malolactic fermentation by Cunha-Silva and coworkers. In most wines, this process takes place up to 40 days, so a biosensor with higher stability was developed. For this purpose, the enzyme LOx was crosslinked with a copper metallic framework on a chitosan layer over a platinum-modified screen-printed electrode. The presence of the metallic framework enhanced the catalytic activity, and hence improved the sensitivity (14.65 µA·mM^−1^) of the device, while the Pt allowed for working at an interference-free potential of 0.15 V. Finally, the layer formed by the biopolymer chitosan and the framework acted as a barrier, delaying the displacement rate of the enzymatic reaction product to the Pt surface, and thus increasing the amperometric resolution up to 50 mM. This barrier avoids the lixiviation of the enzymes and protects them from environmental hazards, lengthening its lifetime to at least 50 days with 100% of response sensitivity. For this reason, the developed device was proposed to monitor the fermentation process of wine. Moreover, it exhibited the highest sensitivity reported among biosensors applied in wines analysis [133].

Equally interesting is the biosensor developed by Vargas et al. for beer analysis. This device is composed of two enzymes—lactate dehydrogenase and diaphorase—coimmobilised with the redox mediator tetrathiafulvalene in a dialysis membrane. The novelty of this research lies in the accomplishment of a dual determination of both isolated lactate isomers. This milestone was achieved in a single experiment without previous separation steps, working with the developed biosensor for D-lactate tracing and with a commercial biosensor for L-lactate. Moreover, they were combined in a semiautomatic flow injection (FI) system and tested, proving inherent advantages of simplicity, sufficient sensitivity and assay time. The proposed FI methodology was compared with a commercial enzymatic kit in the analysis of both isomers in beer samples. The correlation coefficients (0.996) prove the suitability of this device for the quantitation of lactate isomers in beer samples. Moreover, the linear range of the biosensor device (0.5–17 mg·L^−1^ for D-lactic and 0.039–13 mg·L^−1^ for L-lactic) was suitable for the ten-times-diluted samples (with concentrations ranging from 2.6 to 9.9 mg·L^−1^). Therefore, this system can be applied in routine food quality control avoiding expensive columns with chiral stationary phases [125].

The efforts of Shkotova and coworkers in developing an amperometric multibiosensor for quality control of wines should be praised. For this purpose, a gold thin-film-based amperometric multitransducer was modified with a solution of bovine serum albumin and lactate and glucose oxidase. After the deposition of the mixture of each enzyme on the surface of the working electrodes, the device was stored for only 10 min in a crystalliser with an atmosphere of saturated glutaraldehyde vapour. The selectivity of the multibiosensor was proven, testing the response of the device in presence of major components of wine (e.g., glucose or ethanol), and ultimately several wines were analysed with the developed device and with high-performance liquid chromatography to validate the method. There was a good agreement between data from both methods; however, the correlation coefficient for glucose was better (0.998) than the lactate one (0.718) [134]. 

Another significant approach is the development of a biosensor with widespread applicability on different food matrices. In this respect, the advances reported by Bravo et al. are significant. Interestingly, they were able to prepare an electrochemical device using gold nanoparticles as an electrocatalyser. After the deposition of the modifier onto the surface of a screen-printed carbon electrode, LOx enzymes were assembled upon it. The nanoparticles showed an impressive electrocatalytic effect on the oxidation of H_2_O_2_, the product of lactate oxidation by the enzyme. Hence, the sensitivity of the sensor towards lactate was enhanced significantly. Moreover, this device was applied successfully on the analysis of three different samples (wine, beer and yoghurt) without the pretreatment step and with validation from a commercial enzymatic-spectrophotometric assay kit. The recovery values were excellent for the wine (98%), beer (100%) and yoghurt (95%) samples [135].

Uygun and coworkers applied another approach for the development of a potentiometric lactate biosensor. A composite pH membrane selective for hydronium ions was constructed by the combination of hydroquinone and quinone in 1:1 mole ratio. The composite was reinforced with the polymer polyvinylchloride-aminated (PVC-NH_2_) and graphite, and LOx enzymes were crosslinked upon it with glutaraldehyde. This device exhibits a wide linear range between 0.05 and 10 mM and a low limit of detection (20 µM). Moreover, successful analysis of several buttermilks and a pickle juice were carried out with the developed biosensor (recoveries values between 97–108%), which acknowledged the usefulness of the device in food sample analysis [136].

Finally, a device based on the combination of two conductometric biosensors, one monoenzymatic and another bienzymatic, was applied for the determination of total lactate, L- and D-lactate in dairy products. The biosensors were prepared through crosslinking of L-lactate oxidase from *Pediococcus* sp. (LODP) or a combination of LODP and horseradish peroxidase (HRP) at the surface of gold interdigitated microelectrodes using glutaraldehyde vapours. On the one hand, the chiral selectivity of LODP towards L-lactate was lost due to the enzyme crosslinking, rendering the monoenzymatic biosensor suitable for total lactate determination. On the other hand, the addition of HRP modified the chiral selectivity of the LODP, enchaining the response of the bienzymatic biosensor toward D-lactic isomer. The linear response of the LODP biosensors toward L-lactate improved from 100 to 200 µM when HRP was added to the device, whereas sensitivity changed from 1.16 ± 0.04 to 4.18 ± 0.04 µS µM^−1^. Moreover, the limit of detection of the bienzymatic biosensors was 0.05 µM. Thereafter, the suitability of the dual device for dairy product analysis was assessed. Firstly, the selectivity of the LODP/HRP towards L-lactate was studied using glucose, lactose and ascorbic acid. Only the latter produced a 70% change in the signal when a 1:1 molar ratio was utilised. However, the typical concentration of ascorbic acid was 500-fold lower than lactate in these kinds of samples. Then, the device was applied in the analysis of three yoghurts, obtaining better recovery values for L-lactate (91–108%) than for D-lactate (83–114%) [133].

In conclusion, lactate biosensors have a promising future in the food industry due to their applicability in quality control and safety monitoring. However, stronger efforts must be carried out to enable the analysis of complex samples and the real-time evaluation of storage products and manufacturing points.
biosensors-12-00919-t003_Table 3Table 3Lactate biosensors employed for the analysis of food samples.BiosensorElectrodeImmobilisation ProcessLRR (μM)Sens. (µA/mM)LOD (µM)Tr (s)Lifetime (Days)Samp.App.Ref.LOx/3,4DHS–AuNP/SPCESPCE-2.6–8005.12.6-30White wine, yoghurt, beerQuality evaluation[135]laponite/Chit/LOx/GCEGCE-10–70011.413.8430White wine, beer, fermented milk-[134]LOx-PVC-NH_2_-Quinhydrone- GraphiteGraphite-50–10,000-2010-Buttermilk, pickle Juice-[136]LOx-Pt&Pd-Nafion-carbonCarbon-50–800-0.1528WineQuality evaluation[124]LOx–Cu-MOF/Chit/Pt/SPCESPCECrosslinking0.75–100014.650.75-50Red wine, white wineControl of malolactic fermentation[133]PtNPs/GCNF–PEI–GA–LOx–Gly–SPCESPCECovalent10–20000.0256.9-547Wine, cidersAnalysis of lactic acid[135]LODP/HRP-AuIDEAuIDE-0.05–210-0.05-35Yogurt-[133]GOx-LOx-BSA-GA-AuAuCovalent5–10000.75-1085Red wine, white wineQuality evaluation[134]N-eicosane-MWCN-LOx-HRP-GPEGPE-5–2443.470.9665456Beverages, wines, sauces-[136]DLDH/DP-TTF-AuAuEntrapment11–424.0950.34-9BeerSimultaneous determination of lactate enantiomers[125]GA-LDH/AuNPs-ERGO-PAH/SPESPECrosslinking500–3000-1-49Yoghurt, wineQuality evaluation[41]3,4DHS: N,N′-Bis(3,4-dihydroxybenzylidene)-1,2-diaminobenzene Schiff base tetradentate ligand; AuIDE: gold interdigitated microelectrodes; AuNPs: gold nanoparticles; App.: application; BSA: bovine serum albumin; Chit: chitosan; Cu-MOF: copper metallic framework; DLDH: D-lactic acid dehydrogenase; DP: diaphorase; ERGO: electrochemically reduced graphene oxide; GA: glutaraldehyde; GCE: glassy carbon electrode; GCNF: graphitised carbon nanofibres; Gly: glycine; GOx: glucose oxidase; GPE: gold planar electrode; HRP: horseradish peroxidase; LRR: linear response range; LOD: limit of detection; LODP: lactate oxidase from Pediococcus; LOx: lactate oxidase; PAH: poly(allylamine hydrochloride); PEI: polyethyleneimine; Pt&Pd: platinum and palladium nanoparticles; PtNPs: platinum nanoparticles; PVC: polyvinyl chloride; Samp.: samples; Sens.: sensitivity; SPE: screen-printed electrode, SPCE: screen-printed carbon electrode; Tr: response time; TTF: tetrathiafulvalene.


## 5. Conclusions and Future Perspectives

Over the last decade, the development of new trends for lactate monitoring has risen year after year. Even though their importance in the biomedical field was previously known, it has raised their interest in the food control ambit as well. In addition, in both scenarios, the real-time and continuous monitoring with a reliable device is still the main goal to achieve. From the point of view of electrochemical (bio)sensors, a wide range of studies have been published concerning this issue, exposing the relevance of the topic currently. The most popular sensors are based on the amperometric approach, due to its simplicity, sensitivity and its convenience for continuous monitoring. Regarding the resulting devices, a copious number of papers have been published, mainly employing enzymes to supply selectivity to the resulting device. Lactate oxidase and lactate dehydrogenase have been fruitfully exploited in this sense. Remarkably, in order to avoid enzyme drawbacks (e.g., pH and T dependency) other authors have started to explore non-enzymatic approaches based on nanomaterials, MOFs or MIPs. However, other issues such as the mandatory alkaline media in metal oxide nanomaterial-based sensors must be solved before there are clear applications. On the other hand, to fulfil the expectation of the sensors in the biomedical field and in certain areas of agrifood, nondilution sampling requires an enhanced linear range of response, dodging enzyme saturation somehow. Several membranes have been rationally designed to improve this feature using different materials, and in many cases, biocompatible polymers (e.g., chitosan). This approach is also useful to diminish the interferents reaching the electrode surface, hence improving the selectivity and the possible fouling resistance of the device. Unfortunately, real applications of the sensors developed are limited.

It is noteworthy to mention some authors reaching high TRLs but not impacting the market in a meaningful manner. One reason might be the challenging sampling in the biomedical field, with sweat analysis being the most successful, but still restricting the monitoring of the patient to sweating conditions. Alternatively, the low representation of commercial kits for lactate monitoring in food control is surprising. Nonetheless, it should be noted that interest in this field is still rising currently. In fact, not only are factories’ and clinics’ interests on the table, but individual ones as well. Day by day, customers are more interested in their own monitoring, as is reflected by the high amount of smartwatch users nowadays. The vast majority of these devices monitor physical parameters, such as heart rate or even blood pressure. It is only a matter of time that chemical parameters will be monitored as well, with the consequent acquisition of the same target population. Furthermore, commercialisation may have more support provided from the glucose-monitoring devices that are already established in society. The rising visibility of these sensors in the regular lives of diabetic people will surely aid the inclusion of a similar approach for lactate monitoring. It is likely that a similar wearable system will be considered based on the extraction of blood and/or interstitial fluid driving the fluid to a miniaturised electrochemical system. Clinician applications may also follow this approach; nevertheless, in this field, IoT-based paths will be required to fully take advantage of these devices. On the contrary, the application in industry is more flexible and should be studied for each situation. However, the possibilities from IoT must be also strongly considered in this sector. Finally, in order to integrate successful continuous lactate monitoring in society, several aspects should be revised and improved. On the one hand, even though sweat is considered as the main biofluid to analyse in healthcare scenarios, other less-explored fluids are susceptible to be taken in account, such as saliva or interstitial fluid, the latter being a rich source of potential biomarkers. Additionally, miniaturisation is strongly recommended as a way to couple lactate and other analyte sensors in a more efficient and comfortable manner. Last but not least, to step forward and reach a meaningful monitoring, more efforts concerning the inclusion of sensors in the IoT tools of the future are required. Nevertheless, one thing is clear, it is not ambitious to think that no matter the application—either clinical or agrifood—the future of lactate monitoring will advance hand-by-hand with electrochemical (bio)sensors.

## Figures and Tables

**Figure 1 biosensors-12-00919-f001:**
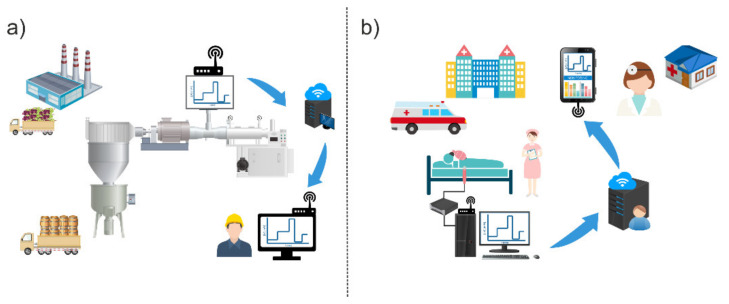
Scheme of relevant high-value situations for real and continuous lactate monitoring, (**a**) wine production and (**b**) intensive care units (ICUs).

**Figure 2 biosensors-12-00919-f002:**
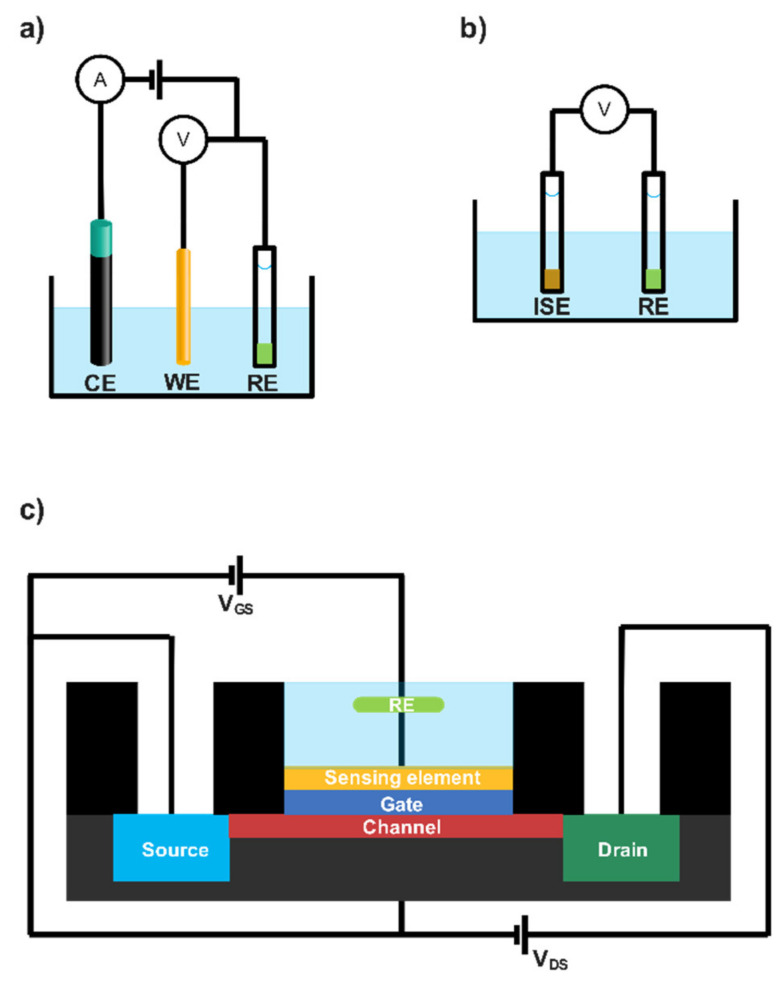
Scheme of main transduction mechanisms for lactate electrochemical (bio)sensors: amperometric (**a**); voltammetric (**b**) and field-effect transistor (**c**).

**Figure 3 biosensors-12-00919-f003:**
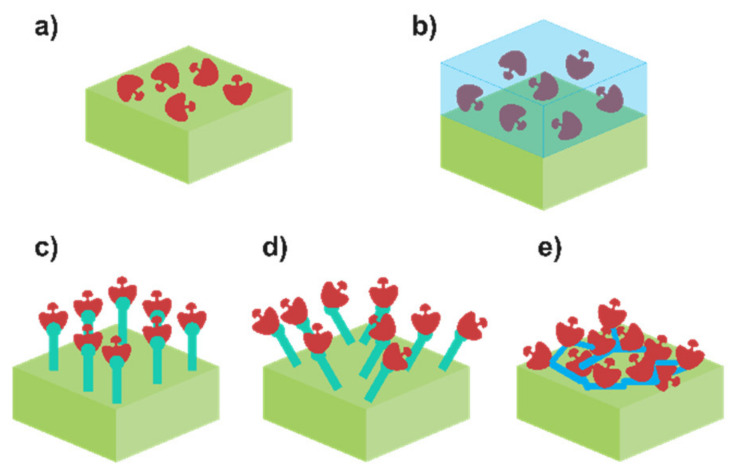
Scheme of different enzyme immobilisation approaches: (**a**) adsorption, (**b**) physical entrapment, (**c**) oriented covalent binding, (**d**) random covalent binding and (**e**) crosslinking methods.

**Figure 4 biosensors-12-00919-f004:**
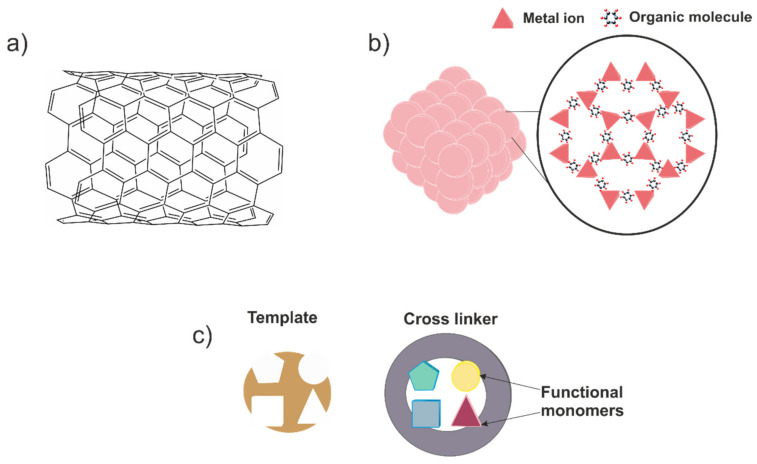
Main alternate approaches to enzyme-based lactate sensors: (**a**) nanomaterials, (**b**) metal–organic frameworks (MOFs) and (**c**) molecularly imprinted polymers (MIPs).

**Figure 5 biosensors-12-00919-f005:**
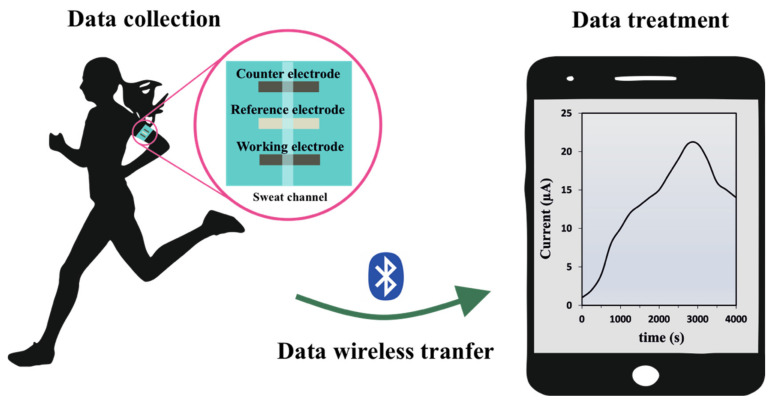
General scheme of the performance of a wireless wearable lactate amperometric biosensor for sweat monitoring.

**Figure 6 biosensors-12-00919-f006:**
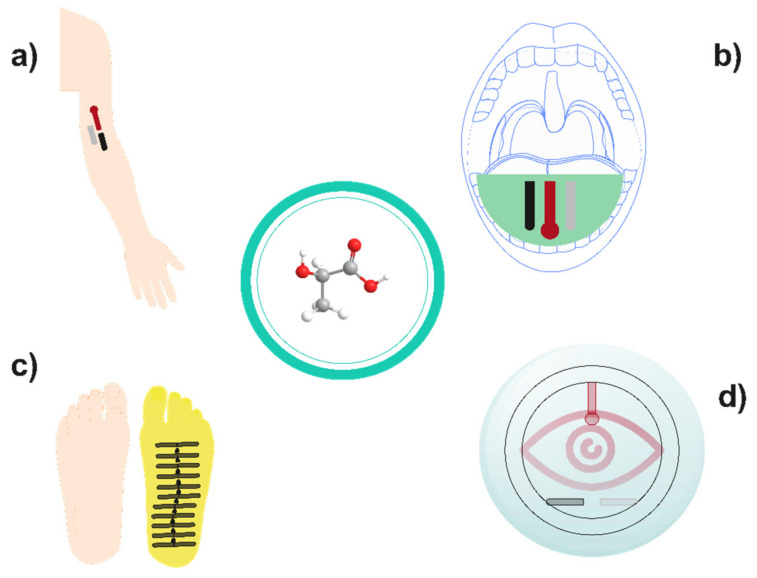
Several wearable lactate sensor examples: epidermal tattoo (**a**) [85], mouthguard (**b**) [84], bioelectronic sock (**c**) [86] and biosensor lens (**d**) [87].

**Figure 7 biosensors-12-00919-f007:**
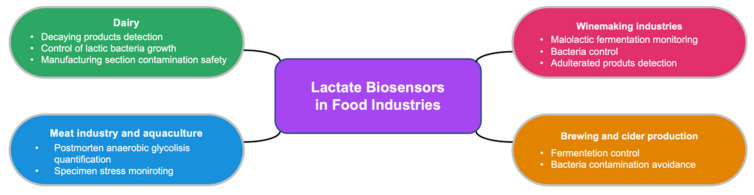
The most eminent applications of lactate biosensors in food industries.

**Table 1 biosensors-12-00919-t001:** Relevant lactate electrochemical (bio)sensors according to their transduction mechanism.

Enzymatic Sensors						
Sensor Architecture	Trans.	LOD (µM)	Sensitivity	LRR (mM)	Sample Applied	Ref
PtE-PDA/PPy/LOx	Amp.	-	37.53 µA mM^−1^ cm^−2^	0–0.5	PBS	[38]
PtµE-poly-m-phenylene diamine/poly(ethylene glycol) diglycidyl ether)-LOx	Amp.	19 ± 7	2.63 ± 0.66 nA mM^−1^	0–1.0	Cerebrospinal fluid in mouse brain	[39]
CE-fSWCNTs/Chit-PBNPs/Chit LOx	Amp.	200	-	1–25	Human sweat	[40]
SPE-ERGO-PAH-AuNPs/LDH-GA	Amp.	1	1.08 µA mM^−1^·cm^−2^	0–3	Wine	[41]
SPCE-PEGDGE/AvLOx	Amp.	25	13 µA mM^−1^ cm^−2^	0–1	PBS	[42]
PtE-poly(phenylenedi- amine)/LOx/glycerol/PVA-SbQ	Amp.	5	204 nA mM^−1^	0.005–1	Blood serum	[43]
Au-PB-Chit/CNTs-LOx-Chit/CNts	Amp.	-	220 nA mM^−1^	0–30	Sweat	[44]
PVC/FC-LDH	Pot.	-	52 mV decade ^−1^	-	Tris buffer	[45]
AuE-LOx/GA/ZnO nanorod	Pot.	0.1	41.33 ± 1.58 mV decade^−1^	0.0001–1	PBS	[46]
Si_3_N_4_-PAA/NHS-EDC/LOx	Pot.	0.0002	49.7 mV decade^−1^	0–0.00005	PBS	[47]
Ti-Au/Nafion/Chit/LDH/GA	GFET	-	-	0–7.5	Human plasma	[48]
Au-Os redox polymer/LOx	OFETs	-	-	0–10	PBS	[49]
Non-enzymatic sensors						
GCE-Nafion/NiO	Amp.	27	62.35 μA mM^−1^ cm^−2^	0.01–7.75	NaOH	[50]
GCE-Nafion/Co_3_O_4_	Volt.	6	-	0.5–3.0	NaOH	[51]
	Amp.	10				
GCE-CuO/MWCNTs/Nafion	Volt.	0.088	633.0 pA mM^−1^ cm^−2^	0.0001–10	Serum samples	[52]
Ti-PTFE/PPy-MWCNTs	Amp.	51	2.9 µAmM^−1^ cm^−2^	1–15	Sweat	[53]
BDD-NiNPs	Amp.	0.72 ± 0.09	(24.70 ± 0.36) μA L C^−1^ mol^−1^	6–120	NaOH	[54]
SPCE-NiCo layered double hydroxide	Amp.	533	30.59 ± 0.34 μA mM^−1^ cm^−2^	5–25	NaOH	[55]
NiF-NiS-NC@NiS-MS	Amp.	0.5	0.39 μA μM^−1^	0.0005–0.085	Urine	[56]
NH_2_-GP-Cu_3_(btc)_2_	Amp.	5	0.029 mA mM^−1^ cm^−2^	0.05–22.6	Sweat	[57]
SPCE-NiCo (layered double hydroxide)	Amp.	400	83.98 μA mM^−1^ cm^−2^	2–26	Sweat	[58]
GCE-rGO-AuNPs-MIP	Amp.	9 × 10^−5^	1.9 × 10^5^ μA L mol^−1^	1 × 10^−7^–1 10^−6^	Sugarcane vinasse	[59]
CE-AgNWs-MIP	Amp.	0.22	4.5 × 10^−6^ A M^−1^	0.001–100	Sweat	[60]
GCE-Nafion/CuO	Amp.	-	80.33 μA mM^−1^	0.01–27.76	-	[61]

Amp.: amperometric; AuE: gold electrode; AvLOx: rationally engineered Aerococcus viridans; BDD: boron-doped diamond; CE: carbon electrode; Chit: chitosan; EDC: 1-ethyl-(3- dimetylaminopropyl)carbodiimide; ERGO: electrochemically reduced graphene oxide; FC: ferrocene; fSWCNTs: functionalised single-walled carbon nanotubes; GA: glutaraldehyde; GCE: glassy carbon electrode; g(CPE): graphite and conductive polyethylene; GFET: common gate graphene-based field-effect transistor; LDH: lactate dehydrogenose; LOD: limit of detection; LOx: lactate oxidase; LRR: linear range response; MIP: molecularly imprinted polymer; MS: microsphere; MWCNTs: multiwalled carbon nanotubes; NC: nanocluster; NH2-GP-Cu3(btc)2: electro catalytic Cu3(btc)2 nanocubes with freestanding amino-functionalised graphene paper; NHS: N-hydroxysuccinimide; NiF: nickel foam; NPs: nanoparticles; NWs: nanowires; OFET: organic field-effect transistor; PAH: poly (allylamine hydrochloride; PB: Prussian blue; PBS: phosphate buffer solution; PDA: polydopamine; PEGDGE: diglycidyl ether; lactate oxidase; Pot.: potentiometric; PPA: polyacrylic acid; PPY: polypyrrole; PtE: platinum electrode; PTFE: polytetrafluoroethylene; PtµE: micro platinum electrode; PVA-SbQ: photopolymer-containing styrylpyridine groups; PVC: polyvinyl chloride; SPCE: screen-printed carbon electrode; SPE: screen-printed electrode; Trans.: transducer; Volt.: voltammetric.

## Data Availability

Not applicable.

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
