# Peer review of "What Is Left for Real-Life Lactate Monitoring? Current Advances in Electrochemical Lactate (Bio)Sensors for Agrifood and Biomedical Applications"

_biosensors, 2022, doi:10.3390/bios12110919_

Round 1
Reviewer 1 Report
The authors present a review of the electrochemical lactate biosensor, a promising tool for food quality control and point-of-care diagnosis. The paper is well written. I just have two concerns about the discussion.
The authors pointed out that the practical applications of electrochemical lactate biosensors are limited. In addition to the authors' discussion, I want to know whether these two points are essential or not.
1, The environment in a factory or customer's home is usually more complicated than in labs, such as the heterogeneous temperature and humidity. Do these environmental factors affect the measurement reliability of electrochemical biosensors?
2, The actual samples, such as serum and food, are usually considered complex media compared with the pure buffer. Are the electrochemical biosensors able to recognize the lactate in complex media? In other words, do they have enough measurement specificity?
Reviewer 2 Report
The authors came up with a good review work in a systematic manner. Still some improvements needs to be done before publication.
1.The authors need to address/add contents related to slefpowered/advanced lactate sensor integrated with IoT/AI wearable or miniaturized systems. There are many reports available addressing this integrations with sensor devices.
2. The conclusion part needs to be revised towards the advancement of the sensor system like commericialization, system integration to make wearables,miniaturized sensor devices etc.
3. The authors should consider adding/summarizing in a paragraph which describe the parameters need to improve the current systems before the conclusion section.
